# Efficacy and Experience of Bacteriophages in Biofilm-Related Infections

**DOI:** 10.3390/antibiotics13020125

**Published:** 2024-01-26

**Authors:** Monica Gordon, Paula Ramirez

**Affiliations:** Critical Care Department, Hospital Universitario y Politécnico la Fe, Av. Vicente Abril Martorell 106, 46026 Valencia, Spain; gordon_mon@gva.es

**Keywords:** biofilm-related infections, phages, devices, phage cocktail

## Abstract

Bacterial infection has always accompanied human beings, causing suffering and death while also contributing to the advancement of medical science. However, the treatment of infections has become more complex in recent times. The increasing resistance of bacterial strains to antibiotics has diminished the effectiveness of the therapeutic arsenal, making it less likely to find the appropriate empiric antibiotic option. Additionally, the development and persistence of bacterial biofilms have become more prevalent, attributed to the greater use of invasive devices that facilitate biofilm formation and the enhanced survival of chronic infection models where biofilm plays a crucial role. Bacteria within biofilms are less susceptible to antibiotics due to physical, chemical, and genetic factors. Bacteriophages, as biological weapons, can overcome both antimicrobial resistance and biofilm protection. In this review, we will analyze the scientific progress achieved in vitro to justify their clinical application. In the absence of scientific evidence, we will compile publications of clinical cases where phages have been used to treat infections related to biofilm. The scientific basis obtained in vitro and the success rate and safety observed in clinical practice should motivate the medical community to conduct clinical trials establishing a protocol for the proper use of bacteriophages.

## 1. Introduction

The rise of antimicrobial resistance poses a major threat to global health. According to a recent analysis [1], the global burden associated with multidrug-resistant bacteria in 2019 was estimated to cause nearly 5 million deaths, with 1.27 million deaths directly attributed to drug resistance. The selection of bacterial strains becoming progressively less susceptible to common antibiotics leads to a reduction in the number of available effective antibiotics, compelling the use of second-line drugs with lower efficacy and a poor safety profile. Moreover, when confronted with an infection caused by multi-resistant bacteria, the likelihood of empirically selecting an appropriate antibiotic decreases drastically, directly, and negatively impacts the patient’s prognosis.

Bacteria exhibit various mechanisms of innate or acquired resistance. While most affect the direct effectiveness of the antibiotic, other mechanisms involve different ways of resisting them, such as bacterial biofilm. Biofilm formation is one of the best-known mechanisms of adaptive antibiotic resistance, responsible for 70% of all microorganism-induced infections, including device-related infections like vascular central catheters, urinary catheters, endotracheal tubes, orthopedic implants, and tissue infections such as chronic otitis or sinusitis, endocarditis, and lung infection in cystic fibrosis [2,3].

A biofilm is defined as a “microbially derived sessile community characterized by cells that are irreversibly attached to a substratum or interface or to each other, are embedded in a matrix of extracellular polymeric substances that they have produced, and exhibit an altered phenotype with respect to growth rate and gene transcription” [4]. Antimicrobial susceptibility differs between planktonic cells and sessile or biofilm-associated cells due to delayed penetration of antimicrobial molecules through the biofilm matrix, slower growth rate of biofilm organisms, lower metabolic activity, greater opportunity for exchange of antibiotic resistance genes, and other physiological changes due to the biofilm mode of growth [4,5]. These changes favor resistance to antimicrobial treatment and facilitate the development of infections that are difficult to eradicate. In this context, it is necessary to search for alternatives to conventional antimicrobial treatment capable of preventing biofilm formation or disrupting mature biofilm.

The earliest studies of phages date back to the early 1900s. In 1915, Frederick W. Twort described the existence of a factor that dissolved bacterial cultures, suggesting it might be an ultra-microscopic virus [6]. Independently, in 1917, Felix D’Hérelle isolated an invisible microbe with an antagonistic property against the bacillus of Shiga from the feces of patients with enteritis. This invisible microbe could never be cultivated in media in the absence of the bacillus, and therefore it had to be an obligate bacteriophage [7]. D’Hérelle and collaborators began to develop therapeutic applications of phage therapy, and in the 1930s and 1940s, many trials were active [8]. The emergence of antibiotics in the 1940s led to a decrease in interest in phage therapy, but the growing problem of multidrug-resistant bacteria has brought attention back to the field of phage therapy. 

Despite the passage of time, phage therapy has not gained global traction, nor does it receive support from the pharmaceutical industry. Consequently, there is a lack of scientific evidence supporting and outlining protocols for the application of phages to challenging-to-treat infections. While the absence of robust scientific evidence persists, it is crucial to update reviews that aggregate in vitro scientific advances and compile accumulated clinical cases to advocate for the use of phages. In this current review, our goal is to present a selection of publications demonstrating clear pathophysiological justification for phage use in biofilm-mediated infections, along with a compilation of the most recently published cases where phages have shown clinical efficacy in situations where antibiotics have failed.

## 2. Bacteriophages

### 2.1. Life Cycle and Host Interactions

Phages are viruses that exclusively infect their target bacterial cells, posing no harm to other microorganisms. More than 5000 phages have been discovered and studied, with most belonging to the class *Caudoviricetes* [9]. Each particle comprises an icosahedral protein capsid enclosing the genetic material (dsDNA, ssDNA, dsRNA, or ssRNA), a spiral contractile sheath, and tail fibers containing the receptor-binding proteins responsible for bacterial host specifity. Depending on the course of infection, phages can be divided into two types: virulent or lytic, or temperate or lysogenic. Some phages produce depolymerase enzymes capable of degrading biofilm extracellular polymeric substances, facilitating phage penetration into the biofilm. Lytic phages also produce endolysins, enzymes that cleave peptidoglycans from the bacterial cell wall. Bacteriophage genome replicates in the bacterial cytoplasm, and, after bacterial cell lysis, new phage virions are released in a quantity (burst size) dependent on the phage, the state of the bacteria host, nutritive compounds surrounding the host, and other environmental factors. Lysogenic phages insert their genome within the bacterial genome and remain in a latent state called prophage, being replicated as part of the bacterial chromosome until the lytic cycle is activated by stress or cellular damage. This process can facilitate the transfer of antibiotic resistance genes but can also be therapeutically used to transfer genes that make bacteria more susceptible to certain antibiotics [10,11,12]. 

### 2.2. Advantages over Antibiotics

Bacteriophages are bactericidal and exhibit activity against multidrug-resistant bacteria, given that their mechanism of action differs from antibiotics. Upon systemic administration, phages are widely distributed, demonstrating their ability to penetrate the blood–brain barrier. They are highly specific, sparing human or animal cells, and do not disturb the natural microbiota in host organisms. The emergence of phage-resistant bacteria is notably slower compared to antibiotic-resistant bacteria. Additionally, phage production is rapid and cost-effective, resulting in lower therapy costs than conventional antibiotic treatments [12,13,14]. 

Regarding biofilm, bacteriophages can penetrate the inner layers, where the high density and proximity of bacteria allow rapid phage replication and the release of new phage particles. Moreover, bacteriophages can infect stationary-phase cells and eliminate them upon reactivation. Bacteriophages also have the ability to produce enzymes inhibiting quorum sensing activity and biofilm production [10,15].

### 2.3. Limitations

Due to the narrow host range of bacteriophages, early, accurate identification of the etiological agent is crucial, unlike antibiotics. Phage replication facilitates the transfer of genetic material between bacteria, including virulence factors and resistance genes. Bacterial lysis also liberates substances such as lipopolysaccharides and endotoxins, which may trigger an inflammatory cascade. Finally, the patient’s immune system can eliminate these viruses from the systemic circulation, diminishing the efficacy of phage therapy [10,16].

The biofilm matrix may impede phage diffusion by containing microbial enzymes that can deactivate phages, dead cells where phages cannot replicate, and components such as lipopolysaccharides and teichoic acids that restrict phage access to biofilm cells. In biofilms, cells in deeper layers exhibit reduced metabolic activity due to oxygen and nutrient deprivation, potentially hindering phage propagation. Phage-resistant phenotypes may emerge within the biofilm as early as 6 h after infection, driven by various mechanisms: prevention of phage genome integration into the bacterial genome, degradation of the phage genome, obstruction of phage replication, transcription, and translation through structural modifications or masking of bacterial receptors. Quorum sensing, the communication system between cells within biofilm through extracellular chemical molecules, can further reduce metabolic activity and regulate antiphage mechanisms [5,10,17].

### 2.4. In Vitro Studies

#### 2.4.1. Single Phages

Several studies have analyzed the antibiofilm activity of single phages in vitro. Variables such as smaller phage genomes, higher burst sizes, shorter phage latent periods, or higher phage concentrations are frequently associated with greater biofilm reduction [15]. The most commonly studied multidrug-resistant bacteria involved in nosocomial infections have been examined.

In a recent study, Adnan M. et al. [18] isolated and characterized bacteriophage MA-1 from wastewater to control biofilm formation by multidrug-resistant *Pseudomonas aeruginosa*-2949. After 6 h of treatment, significant reductions of 2.0, 2.5, and 3.2 folds in 24, 48, and 74-h-old biofilms were observed.

Jamal M. et al. [19] isolated and characterized phage Z, a bacteriophage that inhibits multidrug-resistant *Klebsiella pneumoniae* in planktonic and biofilm form. Reductions of biofilm biomass of 2-fold after 24 h and 3-fold after 48 h were observed. Biofilm cells and stationary-phase bacteria were killed at a lower rate than planktonic bacteria.

Gu Y. et al. [20] used multidrug-resistant uropathogenic *Escherichia coli* MG1655 to isolate and enrich bacteriophages from wastewater. Phage vB_EcoP-EG1 was stored, and after 24 h of treatment, the biofilm biomass of strain MG1655 and clinical strain 390G7 was significantly lower compared to the untreated control (median reduction of 60% and 50%, respectively). 

Lungren M.P. et al. [21] designed an in vitro study to determine the utility of bacteriophages for eliminating biofilm on central venous catheter material. Following 24 h of *Staphylococcus aureus* inoculation, treatment of 10 silicone discs with 10^8^ plaque-forming units/mL solution of *staphylococcal* bacteriophage K during 24 h significantly reduced mean colony-forming units (6.7 × 10^1^ in the experimental arm vs. 6.3 × 10^5^ in the control arm; *p* < 0.0001). 

#### 2.4.2. Phage Cocktail

Phage cocktails are designed to delay the emergence of phage-resistant bacteria and extend the narrow phage host range [5]. Fu et al. [22] investigated the effect of pretreating hydrogel-coated catheters with *Pseudomonas aeruginosa* bacteriophages M4 on biofilm formation. Pretreatment for 2 h at 37 °C reduced the mean viable biofilm count (4.03 log_10_ CFU cm^2^ vs. 6.87 log_10_ CFU cm^2^ on untreated catheters; *p* < 0.001). Regrowth of the biofilm occurred between 24 h and 48 h, and biofilm isolates resistant to phage M4 were recovered from pretreated catheters. Based on the phage susceptibility profiles of these isolates, a five-phage cocktail was developed, and the pretreatment of catheters with this cocktail reduced the 48-h mean biofilm cell density by 99.9%, with fewer biofilm isolates resistant to these phages. 

#### 2.4.3. Combination Therapy

The combination of phages and antibiotics has a synergistic effect on the elimination of biofilm. Ryan et al. [23] identified antimicrobial synergy between bacteriophage T4 and cefotaxime in the in vitro eradication of biofilms of the T4 host strain *Escherichia coli* 11303. The addition of medium (10^4^ PFU/mL) and high (10^7^ PFU/mL) phage titers reduced the minimum biofilm eradication concentration value of cefotaxime from 256 to 128 and 32 µg/mL, respectively. 

Chaudry W. et al. [24] studied the effect of combinations of two phages and five classes of bactericidal antibiotics on the 48-h-old biofilm of *Pseudomonas aeruginosa* PA14. When the phage mixture was added 4 or 24 h before the antibiotics, the best results were obtained. Some antibiotics, such as gentamicin and tobramycin, were more effective at lower doses, and their efficacy considerably increased when the phage was administered before the antibiotics.

### 2.5. Human Studies

Although knowledge of this therapeutic method has existed for over a hundred years, there remains a substantial lack of randomized clinical trials that could, according to current standards, confirm the efficacy of applying bacterial viruses to combat bacterial infections. The reason behind the absence of clinical trials is unclear. The need for specific phages for each bacterium requires the creation of living libraries with the ability to activate the virus and replicate it in real time in order to be a feasible treatment. Methodologically, there would be a need for some empirical agreement on the administration protocol. Finally, it’s possible that there is currently no economically powerful pharmaceutical industry supporting this therapeutic option and conducting relevant clinical trials. However, the results of many published case studies are promising.

In the 1980s, the use of phages or combinations of phages with or without antibiotics was studied in patients with suppurative bacterial infections or chronic urinary tract infections in Poland. More than 90% of the treated patients showed a good response; surprisingly, the rate of clinical cure was lower in those treated with a combination of phages and antibiotics [25,26,27,28,29]. 

Recently, several clinical cases have been published. The most common among the treated patients are lung infections, device-related infections, and chronic wounds. Phage therapy was initiated in the majority of patients after the failure of conventional treatments. However, the rate of improvement or even resolution seems to be as high as 96% in the 76 clinical cases collected in this review.

## 3. Methodology

The pipeline to develop a phage therapy formulation for a biofilm infection usually starts with the isolation of the bacterial pathogen causing the infection, followed by the screening for phage(s) with lytic activity against the isolated strain. These phages can be obtained from phage banks, large phage collections from phage therapy centers, or pharmaceutical companies. This screening identifies the best phage candidates for therapy, typically resulting in the formulation of a phage cocktail—multiple phages in a single preparation. There’s mention of a potential phage adaptation to the biofilm phenotype before phage treatment to enhance therapy outcomes. 

Phages can be administered orally, intravenously, in inhaled formulations, or incorporated into hydrogels or other matrices for efficient delivery in topical applications for chronic wounds or as a coating for implanted materials. However, ensuring the stability of phage formulations is crucial.

The phage therapy protocol should be adjusted according to the type of infection. For example, the preferred routes of phage administration for chronic lung infections are intravenous, inhalation, or the combination of both routes during treatment. In chronic wounds, treatment is usually carried out by topical application. Phage doses vary widely in the literature (between 10^6^ and 10^10^ plaque-forming units (PFU)/mL) and should be adapted to the infection, bacterial inoculum, and response. The number of administrations during the treatment duration varies among different authors [30]. 

Treatment response is assessed through bacterial cultures and monitoring specific infectious symptoms, which can vary according to the type of infection. For instance, in patients with chronic lung infection, important indicators include forced expiratory volume, the need for oxygen support, levels of cough and sputum production, and recurrence of exacerbations, providing insights into the efficacy of phage therapy in this biofilm-related condition [30].

### 3.1. Lung Infections

Most published cases involved exacerbations of chronic pathologies where chronic biofilm-mediated bacterial colonization is typical, specifically in cystic fibrosis and bronchiectasis. These patients often experience multiple infections and have received various antibiotics, leading to the frequent selection of multi-resistant bacteria. The use of phages presents a significant advantage in such cases. *Pseudomonas aeruginosa* was the most commonly implicated microorganism, and treatment typically involved a phage cocktail administered over an extended period. Inhaled administration was used in 63% (n = 17) of the cases, with 40% of the patients (n = 11) not receiving concomitant antibiotic treatment. Of the 27 cases that we have been able to collect from the scientific literature, all but one showed major improvement or resolution of the clinical picture (Table 1) [31,32,33,34,35,36,37,38,39,40,41,42,43,44].

### 3.2. Biofilm-Related Cardio-Vascular Devices

Various cardiac devices have been treated with phages due to their association with complicated infections that are unresponsive to antibiotic treatment. However, the shorter duration of phage treatment highlights the lower complexity of these device-related infections compared to chronic lung pathologies. *Staphylococcus aureus* was the most frequently involved bacteria. While systemic treatment was prevalent, local phage application was nearly standard, and all patients received combined treatment with antibiotics. Once again, of the nine cases collected from the scientific literature, all but one showed major improvement or resolution of the clinical picture (Table 2) [40,45,46,47].

### 3.3. Biofilm-Related Chronic Infections Treated with Phages

Chronic wounds (ulcers) and osteomyelitis are the most common chronic clinical conditions treated with phages. The etiology was extremely diverse, often involving multiple bacteria (polymicrobial infection). Antibiotic resistance challenges were frequent, even with limited antibiotic options. Phages were primarily administered locally, mostly in combination with antibiotics and surgery. Resolution was the most frequent clinical outcome, with only one patient among the collected 31 presenting clinical failure (Table 3) [44,48,49,50,51,52,53,54,55,56].

### 3.4. Prosthetic Joint Infections Treated with Phages

Nine cases of prosthetic joint infection have been recently published, all of them occurring in patients older than 60 years. Gram-positive cocci were the predominant etiology, with phage treatment duration not well reported in many cases. All patients underwent surgery, and the infection was resolved without adverse events (Table 4) [57,58,59,60,61,62,63].

## 4. Conclusions

All in vitro data, including animal models, indicate that phages are an optimal therapeutic tool for infections related to the presence of biofilm. Beyond the advantages of phages, such as exclusivity for their bacterial target, effectiveness, and absence of adverse effects, these viruses effectively evade the defense mechanisms of biofilm. To this fundamental aspect, we must add the ability to treat bacteria with resistance mechanisms against common antibiotics.

However, as with other clinical aspects of phage use, the challenge arises when seeking scientific evidence. Clinical trials are lacking, and there are no protocols for indications, duration of use, dosage, or posology. Despite these pragmatic challenges, the number of successful cases published continues to increase. In conclusion, there is an urgent need for systematic studies and validated therapeutic protocols to establish phage treatment in routine clinical practice.

## Figures and Tables

**Table 1 antibiotics-13-00125-t001:** Biofilm-related lung infections treated with phages.

Ref.	Infection	N°	Age (Years)	Etiology	Phage Preparation	Phage Source	Phage Dose (PFU/mL)	Treatment Duration (Days)	Route of Administration	Antibiotics	Clinical Outcome	Adverse Events
[31]	Cystic fibrosis	1	26	*MDR* *P. aeruginosa*	Phage cocktail	AmpliPhi Biosciences	NR	56	Intravenous	Yes	Improvement	No
[32]	Cystic fibrosis	1	17	*MDR* *A. xylosoxidans*	Phage Cocktail	Eliava Institute	1.5 × 10^8^	80	Oral + inhaled	Yes	Improvement	NR
[33]	Necrotizing pneumonia	1	77	*MDR* *P. aeruginosa*	Phage cocktail	AmpliPhi Biosciences	1.0 × 10^9^	7	Intravenous + inhaled	Yes	Resolution	No
[34]	Cystic fibrosis	4	16–38	*MDR* *P. aeruginosa*	Single phage	NR	3.0 × 10^8^	10	Inhaled	No	Improvement	No
[34]	Bronchiectasis	2	71–72	*MDR* *P. aeruginosa*	Single phage	NR	3.0 × 10^8^	10	Inhaled	No	Improvement	No
[35]	Pneumonia-Lung transplant	1	67	*MDR* *P. aeruginosa*	Phage Cocktail	AmpliPhi Biosciences, Naval Medical Research Center	1.0 × 10^9^	134	Intravenous + inhaled	Yes	Improvement	No
[35]	Bronchiectasis-Lung transplant	1	57	*MDR* *P. aeruginosa*	Phage Cocktail	AmpliPhi Biosciences	4.0 × 10^9^	28	Intravenous	Yes	Improvement	No
[35]	Cystic fibrosis-Lung transplant	1	28	*MDR* *B. dolosa*	Single Phage	Adaptive Phage Therapeutics	3.5 × 10^7^	42	Intravenous	Yes	Improvement	No
[36]	Cystic fibrosis-Lung transplant	1	15	*MDR* *M. abscessus*	Phage Cocktail	Pittsburgh Bacteriophage Institute	1.0 × 10^9^	224	Intravenous	Yes	Improvement	No
[37]	Cystic fibrosis	1	26	*MDR* *P. aeruginosa*	Phage Cocktail	AmpliPhi Biosciences	8.0 × 10^8^	56	Intravenous	Yes	Resolution	No
[38]	Cystic fibrosis	4	NR	*MDR* *P. aeruginosa*	Single phage	Yale University	NR	10	Inhaled	No	Improvement	No
[39]	Cystic fibrosis	1	10	*PDR Achromobacter* spp.	Single phage	Adaptive Phage Therapeutics	NR	14	Intravenous	Yes	Improvement	No
[40]	Pneumonia-Heart transplant	1	40	*PDR* *K. pneumoniae*	Phage cocktail	Gabrichevsky Institute	1.0 × 10^8^	4	Inhaled + intra-abdominal	Yes	Resolution	No
[40]	Pneumonia-Lung transplant	1	13	*P. aeruginosa*	Phage cocktail	Gabrichevsky Institute	4.0 × 10^10^	NA	Local	Yes	Resolution	No
[41]	Bronchiectasis	1	81	*M. abscessus*	Phage Cocktail	Pittsburgh Bacteriophage Institute	1.0 × 10^9^	180	Intravenous	Yes	Failure	No
[42]	Cystic fibrosis-Lung transplant	1	12	*PDR* *A. xylosoxidans*	Phage Cocktail	DSMZ collection	5.0 × 10^9^	16	Local + inhaled	Yes	Resolution	NR
[43]	Pneumonia-COPD	1	88	*MDR* *A. baumannii*	Single Phage	Shenzhen Institutes of Advanced Technology	5 × 10–5 × 10^10^	16	Inhaled	Yes	Resolution	No
[44]	Cystic fibrosis	1	43	*P. aeruginosa*	Phage Cocktail	Eliava Institute	9 × 10^6^–1 × 10^7^	1490	Oral + inhaled	Yes	Improvement	No
[44]	Bronchiectasis	1	64	*P. aeruginosa*	Single Phage	Eliava Institute	4 × 10^6^–6 × 10^6^	1095	Oral	No	Improvement	No

MDR—Multidrug-resistant; NR—Not reported; PFU—Plaque-forming unit; COPD—Chronic obstructive pulmonary disease.

**Table 2 antibiotics-13-00125-t002:** Biofilm-related cardiovascular device infections treated with phages.

Ref.	Device	N°	Age (Years)	Etiology	Phage Preparation	Phage Source	Phage Dose (PFU/mL)	Treatment Duration (Days)	Route of Administration	Antibiotics	Clinical Outcome	Adverse Events
[45]	Vascular graft	1	76	*P. aeruginosa*	Single Phage	Yale University	1.0 × 10^7^	1	Local	Yes	Resolution	No
[46]	Left ventricular assist device	1	65	*S. aureus*	Phage Cocktail	AmpliPhi Biosciences	3.0 × 10^9^	28	Intravenous	Yes	Resolution	No
[47]	Prosthetic valve	1	65	*S. aureus*	Phage cocktail	AmpliPhi Biosciences	1.0 × 10^9^	14	Intravenous	Yes	Improvement	No
[40]	Vascular graft	1	52	*P*. *aeruginosa +* *S*. *aureus + E*. *faecium*	Phage cocktail	Gabrichevsky Institute	1.0 × 10^8^	1	Local + oral	Yes	Resolution	No
[40]	Vascular graft	1	59	*S. aureus*	Single Phage	Gabrichevsky Institute	1.0 × 10^9^	2	Local + oral	Yes	Resolution	No
[40]	Left ventricular assist device	1	62	*S. aureus*	Single Phage	Gabrichevsky Institute	1.0 × 10^9^	2	Local + oral	Yes	Resolution	No
[40]	Left ventricular assist device	1	51	*S. aureus*	Phage cocktail	Gabrichevsky Institute	1.0 × 10^9^	8	Local + oral + inhaled	Yes	Failure	No
[40]	Treprostinil pump	1	45	*S. aureus*	Single Phage	Gabrichevsky Institute	4.0 × 10^10^	1	Local	Yes	Resolution	No
[40]	Prosthetic valve	1	66	*E. coli*	Phage cocktail	Gabrichevsky Institute	4.0 × 10^10^	1	Local	Yes	Resolution	No

NR—Not reported; PFU—Plaque-forming unit.

**Table 3 antibiotics-13-00125-t003:** Biofilm-related chronic infections treated with phages.

Ref.	Infection	N°	Age	Bacterial Pathogen	Phage Preparation	Phage Source	Phage Dose (PFU/mL)	Treatment Duration (Days)	Route of Administration	Combined Therapy	Clinical Outcome	Adverse Events
[48]	Osteomyelitis and joint infection	1	60	*XDR P. aeruginosa*	Phage Cocktail	Pherecydes Pharma	4.0 × 10^8^	12	Local	Surgery + antibiotics	Resolution	No
[49]	Chronic wound + osteomyelitis	1	63	*MSSA*	Single Phage	Eliava Institute	NR	49	Local	Antibiotics	Resolution	NR
[50]	Osteomyelitis (pelvis)	1	NR	*P. aeruginosa +* *S. epidermidis*	Phage cocktail	Queen Astrid Military Hospital	1.0 × 10^7^	10	Local	Surgery + antibiotics	Resolution	No
[50]	Osteomyelitis (femur)	1	NR	*XDR P. aeruginosa +* *S. epidermidis*	Phage cocktail	Queen Astrid Military Hospital	1.0 × 10^7^	7	Local	Surgery + antibiotics	Resolution	No
[50]	Osteomyelitis (femur)	1	NR	*S. aureus +* *S. agalactiae*	Phage cocktail	Queen Astrid Military Hospital	1.0 × 10^7^	9	Local	Surgery + antibiotics	Resolution	No
[50]	Osteomyelitis (femur)	1	NR	*E. faecalis*	Phage cocktail	Eliava Institute	1.0 × 10^7^	7	Local	Surgery + antibiotics	Resolution	Yes
[51]	Chronic wound (ulcer)	5	NR	*S. aureus*	Phage cocktail	Banaras Hindu University	1.0 × 10^9^	13	Local	None	Resolution	No
[51]	Chronic wound	6	NR	*E. coli*	Phage cocktail	Banaras Hindu University	1.0 × 10^9^	13	Local	None	Resolution	No
[51]	Chronic wound	9	NR	*P. aeruginosa*	Phage cocktail	Banaras Hindu University	1.0 × 10^9^	13	Local	None	Resolution	No
[52]	Chronic urinary tract infection	1	58	*MDR K. pneumoniae*	NR	Eliava Institute	NR	NA	Oral + local	Antibiotics	Resolution	No
[53]	Osteomyelitis (tibia)	1	42	*XDR A. baumannii + MDR K. pneumoniae*	Phage Cocktail	Naval Medical Research Center and Adaptive Phage Therapeutics	5.0 × 10^7^	11	Intravenous	Antibiotics	Resolution	No
[54]	Chronic urinary tract infection	1	63	*XDR Klebsiella pneumoniae*	Phage Cocktail	Shanghai Institute of Phage	5.0 × 10^8^	5	Local	Antibiotics	Resolution	No
[55]	Chronic bacterial prostatitis	1	33	*MRSA +* *S. haemolyticus+* *E. faecalis + S. mitis*	Phage Cocktail	Eliava Institute	1 × 10^5^–1 × 10^7^	NA	Oral + intra-rectal + intra-urethral	None	Resolution	No
[56]	Osteomyelitis (pelvic bone allograft)	1	13	*MSSA + P*. *mirabilis +* *F*. *magna + C*. *hathewayi*	Phage Cocktail	Queen Astrid Military Hospital	1.0 × 10^7^	14	Local	Surgery + antibiotics	Improvement	NR
[44]	Chronic urinary tract infection	1	72	*MDR K. pneumoniae*	Phage Cocktail	Eliava Institute	8 × 10^6^, 7 × 10^8^	365	Oral + local	Antibiotics	Failure	NR

MDR—Multidrug-resistant; MSSA—Methicillin-sensitive *Staphylococcus aureus*; MRSA—Methicillin-resistant *Staphylococcus aureus*; NR—Not reported; PFU—Plaque-forming unit; PDR—Pandrug-resistant; XDR—Extensively drug-resistant.

**Table 4 antibiotics-13-00125-t004:** Prosthetic joint infection treated with phages.

Ref.	Prosthetic Infection Location	N°	Age	Bacterial Pathogen	Phage Preparation	Phage Source	Phage Dose (PFU/mL)	Treatment Duration	Route of Administration	Combined Therapy	Clinical Outcome	Adverse Events
[57]	Knee	1	80	*MDR* *P. aeruginosa*	Single phage	Eliava Institute	1.0 × 10^8^	5	Local	Surgery and Antibiotics	Resolution	No
[58]	Knee	1	72	*MRSA*	Single Phage	Adaptive Phage Therapeutics	5.4 × 10^9^	3	Local + intravenous	Surgery and antibiotics	Resolution	Yes
[59]	Knee	1	49	*MSSA*	Phage Cocktail	Pherecydes Pharma	1.0 × 10^10^	NR	Local	Surgery and antibiotics	Improvement	No
[60]	Knee	1	80	*MSSA*	Phage Cocktail	Pherecydes Pharma	1.0 × 10^9^	NR	Local	Surgery and antibiotics	Resolution	NR
[60]	Knee	1	84	*MSSA*	Phage Cocktail	Pherecydes Pharma	1.0 × 10^9^	NR	Local	Surgery and antibiotics	Resolution	NR
[60]	Knee	1	83	*MSSA*	Phage Cocktail	Pherecydes Pharma	1.0 × 10^9^	NR	Local	Surgery and antibiotics	Improvement	NR
[61]	Knee	1	88	*P. aeruginosa*	Phage Cocktail	Pherecydes Pharma	1.0 × 10^9^	NR	Local	Surgery and Antibiotics	Resolution	NR
[62]	Knee	1	61	*MSSA*	Single Phage	Adaptive Phage Therapeutics	2.9 × 10^10^	42	Local + intravenous	Surgery and Antibiotics	Resolution	No
[63]	Knee	1	79	*MDR* *S. epidermidis*	Single Phage	PhagoMed	2.0 × 10^10^	NR	Local	Surgery and Antibiotics	Resolution	No

MDR—Multidrug-resistant; MSSA—Methicillin-sensitive *Staphylococcus aureus*; MRSA—Methicillin-resistant *Staphylococcus aureus*; NR—Not reported; PFU—Plaque-forming unit.

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
