# Peer review of "Efficacy and Experience of Bacteriophages in Biofilm-Related Infections"

_antibiotics, 2024, doi:10.3390/antibiotics13020125_

Round 1
Reviewer 1 Report
Comments and Suggestions for Authors
The manuscript “Efficacy and experience of bacteriophages in biofilm related infections” is well organized and carefully researched. The data is presented in an easy-to-read form and references are able to be located without too much issue. The authors have composed a well-crafted scientific manuscript that I’m certain will be referenced numerous times in the future.
There are only a few minor items to check, I’ve listed them below. Also, it would be a good idea to have a once over review for grammar and usage just to be sure. Otherwise, great paper.
Line 163, “More tan 90%” perhaps meant to be “than.”
The References section needs some attention to detail. The citations are not all the identical format.
Some have the year in parentheses after the authors, some with the journal and pages.
Some author name initials have a period, others do not.
There is some capitalization in ref 34, 38 that seems awkward.
References overall just need a through double check to make sure all is in order.
Author Response
Dear editor, Thank you very much for your comments. There were indeed errors in the text, such as the example mentioned above, we have corrected this error and revised the rest of the text.
We have also reviewed all the references that, as the reviewer had found, had discrepancies in their presentation.
We attach the manuscript with all the corrections.

Reviewer 2 Report
Comments and Suggestions for Authors
The paper explores the promising potential of phages as an optimal therapeutic tool for infections associated with biofilm presence, backed by in vitro data, including animal models. It underscores the advantages of phages, such as their bacterial target exclusivity, efficacy, and absence of adverse effects, along with their ability to evade biofilm defense mechanisms. The paper further highlights the prospect of treating bacteria with resistance mechanisms against common antibiotics. However, it draws attention to a critical issue - the lack of scientific evidence in clinical trials and established protocols for phage use. The paper mentions the absence of guidelines regarding indications, timing, dosage, and posology, revealing a pragmatic challenge in the utilization of phage therapy. Despite these challenges, the authors note a growing number of successful cases published, emphasizing the need for systematic studies to validate therapeutic protocols and integrate phage treatment into routine clinical practice. In my opinion, it is recommended to enhance the clarity of the abstract by specifying the types of infections and biofilms under consideration. Furthermore, the introduction can be strengthened by providing a more detailed background on the limitations of current antibiotic treatments and the rising concerns about antibiotic resistance, which phages aim to address, before literature review. Additionally, the abstract could benefit from a concise statement summarizing the key findings of the review. Furthermore, it is suggested to elaborate on potential reasons behind the lack of clinical trials and protocols, addressing possible barriers and proposing avenues for future research. This would contribute to a more comprehensive understanding of the challenges associated with implementing phage therapy in routine clinical practice. Overall, the review paper provides a promising foundation but would benefit from refining the abstract, strengthening the introduction, and providing more in-depth insights as a whole to contribute meaningfully to the current knowledge in the field. Below authors can also find other advice that can help improve their manuscript.
#1 There are several robust typo and grammatical mistakes which should be fixed. Please check the manuscript as a whole, e.g., unnecessary commas or missing commas and other minor mistakes. Look some important mistakes: change “[…] including device-related infections (vascular central catheters, urinary catheters, endotracheal tubes, orthopedic implants) […]” to ““[…] including device‒related infections, e.g., vascular central catheters, urinary catheters, endotracheal tubes, and orthopedic implants, […]”. Please check all manuscript.
#2 Please add more quantitative information in the abstract. I recommend authors use the following reference to adjust their abstract (https://doi.org/10.1016/j.carbon.2007.07.009).
#3 A robustly paragraph about previous reviews must be added in the introduction section in order to clarify the need to study this topic based on the weak literature.
#4 The novelty of the review is not explicitly expressed. What aspects does this review bring that make it superior to others?
Comments on the Quality of English LanguagePlease see report!
Author Response
Dear editor,
Thank you very much for your comments. Indeed, we had made a serious mistake with the abstract since we had mistakenly repeated the conclusions section. We hope that the current format of the asbtract meets the stated and recommended expectations.
We have changed the introduction and added several paragraphs (in yellow) following the editor's recommendations
The entire text has been reviewed and we have requested external help to ensure the absence of errors due to language
Regarding the paragraph commenting on previous revisions and the objectives of our manuscript, we have added a new paragraph.
We attach the new version of the article

Reviewer 3 Report
Comments and Suggestions for Authors
Dear Editor
The manuscript "Efficacy and experience of bacteriophages in biofilm related infections" is unable to provide the significant information as outlined in the title. There is much potential in this domain, however, the Authors have ignored critical analysis of previous data to recommend or suggest further studies in this domain.
The introduction does not provide critical analysis if the domain.
There are several points that are cited in the text without any references.
Data outlined in Table 1 and 2 is presented as simple summary.
The abstract does not provide sufficient information/ hypothesis/ need of the review, further conclusion section is similar to the abstract.
Thanks and Regards
The abstract does not provide any significant information or unable to construct hypothesis of the manuscript, further it is exactly similar as "Conclusion".
Comments on the Quality of English LanguageThe English is good to understand.
Author Response
Dear Reviewers
We regret that our manuscript did not meet the expectations created. However, we have carried out a review of the most recent literature on the topic and we believe we have included all the most relevant aspects on the topic. It has not been possible to include all the published works given the high number, it has been necessary to restrict ourselves to the most current ones (which in turn refer to previous ones), to a limited number that is representative of each aspect and those of the highest scientific quality. . We do not believe we have ignored critical works on the reviewed topic.
We have improved the introduction to try to present the topic with greater clarity and intensity.
We have placed the references at the end of each paragraph if several of the sentences that made it up came from the same source
The data from the clinical cases in the tables are descriptive because there is no other way to present them. The text attempts to summarize the most relevant aspects of the set.
We have made a mistake in the initial abstract since the conclusions section was copied by mistake as the reviewer has detected. This error has been corrected
There are no acknowledgments to mention in this review article.
We attach a new version of the manuscript and hope to have addressed some of the reviewer's concerns.

Round 2
Reviewer 3 Report
Comments and Suggestions for Authors
Dear Editor
The manuscript "Efficacy and experience of bacteriophages in biofilm related infections" has been significantly improved, thanks to the Authors for their improved work.
Thanks and Regards
Comments on the Quality of English LanguageThe English language is good.